# Zimin patterns in genomes

Nikol Chantzi[1,2], Ioannis Mouratidis[1,2], Ilias Georgakopoulos-Soares[1,2*]

**1** Institute for Personalized Medicine, Department of Biochemistry and Molecular Biology, The Pennsylvania State University College of Medicine, Hershey, Pennsylvania, United States of America, **2** Division of Pharmacology and Toxicology, College of Pharmacy, The University of Texas at Austin, Dell Pediatric Research Institute, Austin, Texas, United States of America

* ilias@austin.utexas.edu

## Abstract

Zimin words are words that have the same prefix and suffix. They are unavoidable patterns, with all sufficiently large strings encompassing them. Here, we examine for the first time the presence of k-mers not containing any Zimin patterns, defined hereafter as Zimin avoidmers, in the human genome. We report that in the reference human genome all k-mers above 104 base-pairs contain Zimin words. We find that Zimin avoidmers are most enriched in coding and Human Satellite 1 regions in the human genome. Zimin avoidmers display a depletion of germline insertions and deletions relative to surrounding genomic areas. We also apply our methodology in the genomes of another eight model organisms from all three domains of life, finding large differences in their Zimin avoidmer frequencies and their genomic localization preferences. We observe that Zimin avoidmers exhibit the highest genomic density in prokaryotic organisms, with *E. coli* showing particularly high levels, while the lowest density is found in eukaryotic organisms, with *D. rerio* having the lowest. Among the studied genomes the longest k-mer length at which Zimin avoidmers are observed is that of S. *cerevisiae* at k-mer length of 115 base-pairs. We conclude that Zimin avoidmers display inhomogeneous distributions in organismal genomes, have intricate properties including lower insertion and deletion rates, and disappear faster than the theoretical expected k-mer length, across the organismal genomes studied.

## Author summary

In this study, we investigate a special type of DNA sequence that we call "Zimin avoidmers." These are sequences that possess a unique property: they avoid a specific kind of self-embedded repetition known as a Zimin pattern. Because they lack this repeated structure, they function as an anti-pattern within the genome. This is particularly intriguing, as a known theorem guarantees that any sufficiently long DNA sequence must contain Zimin patterns. With this in mind, our goal is to characterize how often these pattern-free sequences appear,

**Data availability statement:** All code is released under the MIT License. All the source data and the scripts in this study can be found in the Zenodo repository (https://zenodo.org/records/17774048) and at https://github.com/Georgakopoulos-Soares-lab/Avoidmers-DNA. The eukaryotic model organisms were downloaded directly from UCSC genome browser at https://hgdownload.gi.ucsc.edu/downloads.html (Perez et al. 2025). The bacterial reference genomes GCF_000008865.2, GCF_000013425.1, GCF_000240185.1 were downloaded separately from NCBI RefSeq database [44] at https://www.ncbi.nlm.nih.gov/refseq/ [45]. The accession number GCF_000008865.2 can be found here: https://www.ncbi.nlm.nih.gov/datasets/genome/GCF_000008865.2/ Or here: https://ftp.ncbi.nlm.nih.gov/genomes/all/GCF/000/008/865/GCF_000008865.2_ASM886v2/ .The high complexity regions were obtained from Pirogov et al. http://guanine.evolbio.mpg.de/complexity/ (Pirogov et al. 2021). The VCF file of the germline mutations for the human pangenome was obtained from the Human Pangenome Reference Consortium (HPRC) (https://s3-us-west-2.amazonaws.com/human-pangenomics/pangenomes/freeze/freeze1/minigraph-cactus/hprc-v1.1-mc-chm13/hprc-v1.1-mc-chm13.vcfbub.a100k.wave.vcf.gz) [46]. Finally, the mutation rates in human genomes were downloaded from http://genetics.bwh.harvard.edu/downloads/Vova/Roulette/ (Seplyarskiy et al. 2023).

**Funding:** This work was supported by the National Institute of General Medical Sciences of the National Institutes of Health (R35GM155468 to I.G.S.). The funders had no role in study design, data collection and analysis, decision to publish, or preparation of the manuscript.

**Competing interests:** The authors have declared that no competing interests exist.

as well as to determine the maximum lengths they can reach in real genomes across both eukaryotic and prokaryotic organisms. We believe this framework offers a new lens through which to examine genome structure, and it may also prove useful for assessing the validity and behavior of synthetic genomes.

## Introduction

Examining patterns that are absent from the genome space of an organism can reveal valuable biological knowledge including insights into sequence composition and evolution, genomic constraints, and pathogenicity as well as hints regarding the underlying biological mechanisms. Previous work has showcased the shortest k-mer sequences that are absent from a genome, termed nullomers [1]. Another related concept is minimal absent words, which are nullomers, but removing either their leftmost or rightmost character generates a k-mer that is present in the genome [2–4]. Nullomers and minimal absent words have since been used in a plethora of applications, including cancer detection, pathogen surveillance, phylogenetic tree construction, drug development and in forensic science among others [5–12].

A number of research papers have successfully applied theoretical mathematical concepts in genomics, such as the golden ratio, fractals and the Fibonnaci numbers among others [13–15]. Of high interest is the field of combinatorics as its theory and concepts can be applied in genomics in k-mer based investigations. A sesquipower or Zimin word is a sequence of characters over an alphabet with identical prefix and suffix, a concept that was proposed by A. I. Zimin over four decades ago [16]. Zimin words are unavoidable patterns, and all sufficiently long strings must contain Zimin words. The theoretical field of Zimin words has grown considerably since its inception and contains a sizable research literature [17–20]. Repetitive elements are prevalent in organismal genomes, with over half of the human genome composed of repetitive elements, including transposable elements, tandem repeats, and low-complexity regions [21]. Repetitive sequences play key roles in gene regulation, chromatin organization, and genome instability [22]. However, the mathematical principles that govern these repetitions and sequence structure remain incompletely understood. Zimin words are recursive, self-embedding patterns that are mathematically unavoidable in sufficiently long strings, and offer a theoretical lens through which the presence and organization of repetition in genomes can be studied. Despite their rich theoretical foundation in combinatorics on words, Zimin patterns have not been examined in biological sequences. We hypothesized that the distribution and avoidance of Zimin patterns in genomic sequences reflect fundamental constraints on genome structure, function, and evolution. Specifically, we propose that regions avoiding Zimin patterns, termed Zimin avoidmers, may serve as indicators of sequence complexity. By systematically characterizing Zimin words and avoidmers in the human genome and across model organisms, we aim to uncover new principles underlying genome organization and the limits of repeat avoidance.

Here, we identify Zimin words, from which we derive Zimin avoiding sequences, across nine organismal genomes. In this context, we will refer to Zimin avoiding sequences as $Z_3$ *avoidmers*, implying exclusive $Z_3$ avoidance, i.e., the k-mers that avoid, or, equivalently, do not contain any subsequence which captures the *abacaba* pattern on the four letter nucleotide alphabet {*a, g, c, t*} (see Definitions). We report that the upper k-mer limit after which, $Z_3$ Zimin avoiding sequences cannot be detected in the reference human genome is 105 base-pairs (bps). We also show that Zimin avoidmers are less frequent than expected. Zimin avoiding sequences are enriched in coding regions and classic human satellite hsat1B regions in the human genome and are depleted for germline insertion and deletion (indel) variants relative to their surrounding regions. This study provides the first biological application of Zimin words. Future work is required to further investigate their properties, potential roles in organismal genomes and their usage in the development of novel tools and applications.

## Definitions

An *alphabet L* is a collection of symbols often referred to as *letters*. Let $L = \{a_1, ..., a_m\}$ an alphabet of *m* letters. A *word w* is an ordered sequence of letters drawn from the alphabet *L*. We denote by $L^*$ the set of all words over *L*. For any word $w \in L^*$ the length of *w* is denoted by |*w*|. Additionally, we denote by $\varepsilon$ the *empty word*. By definition, $|\varepsilon| = 0$.

In genomics, the nucleotide alphabet is defined naturally as the set of all four nucleotides, i.e., $L = \{a, g, c, t\}$. Small finite words of the nucleotide alphabet of length k are often referred to as *k-mers*. For instance, $w = aagtaag$ is a 7-mer.

A *homomorphism* $\varphi : L_1^* \to L_2^*$ is a function between two non-empty sets $L_1^*$ and $L_2^*$ such that:

$$\varphi(xy) = \varphi(x)\varphi(y),$$

for any words $x, y \in L_1^*$. A homomorphism $\varphi$ is said to be *non-erasing*, if $\varphi(x) \neq \varepsilon$ for any $x \in L_1^*$.

Building on the foundation from [23], let $V = \{v_1, v_2, ...., v_n\}$ a finite set of variables. A *pattern P* is a finite word over *V*. We will say that a word w encounters pattern P, if there is a subsequence z of w and a non-erasing homomorphism φ such that $\varphi(P) = z$.

For example, consider a set of pattern variables $V = \{x, y\}$ and the nucleotide alphabet $L = \{a, g, c, t\}$. The sequence $w = aagtaagaag$ encounters the pattern $Q = xyxx$, since there exists a non-erasing homomorphism $\varphi : V^* \to L^*$ defined by $x \to aag$ and $y \to t$, which embeds pattern *xyxx* into w, since

$$\varphi(Q) = \varphi(xyxx) = \varphi(x)\varphi(y)\varphi(x)\varphi(x) = aagtaagaag = w.$$

## Definition of unavoidability

A pattern *P* is q-*unavoidable* if all but finitely many finite words over an alphabet of size q, encounter *P*. A pattern *P* is unavoidable if it is *q-unavoidable* $q \geq 1$.

For instance, the pattern $P = xx$ is unavoidable in the binary alphabet $q = 2$, since all binary words with at least 4 letters inevitably encounter it.

## Definition of Zimin words in DNA

Since there are four nucleotides in DNA, it is natural to shift our attention primarily to alphabets with four letters and, in particular, the nucleotide alphabet:

$$L = \{a, g, c, t\}.$$

Additionally, we will limit our attention to patterns that are exclusively 4-unavoidable.

For all $n \geq 1$, the $n$-th *Zimin pattern*, is defined recursively, as follows:

$$Z_0 = \varepsilon$$

$$Z_n = Z_{n-1} a_n Z_{n-1}, \ n \geq 1,$$

where $a_n$ denotes a distinct pattern variable. For instance, the first four Zimin patterns are defined as follows: $Z_1 = a$, $Z_2 = aba$, $Z_3 = abacaba$ $Z_4 = abacabadabacaba$.

## Theorem [16,24]

A pattern $P$ containing $n$ different variables is unavoidable if, and only if, $Z_n$ encounters $P$.

The seminal theorem by Zimin [16] has a corollary that over the nucleotide alphabet, any pattern of length n is unavoidable, i.e., any subsequence of sufficiently long k-mer will contain it, if and only if that pattern encounters $Z_n$. Structurally, this means that these patterns are bound to (re)occur within sufficiently long sequences. This triggered a whole new area of research to estimate asymptotically the upper and lower limits of the length of such sequences. Naturally, one would expect due to the repetitiveness of the human genome, such patterns to be rather scarce when compared to genomes that are less repetitive. Another important remark would be that:

In particular, a pattern $P$ containing three different variables *a,b,c* is unavoidable if, and only if, $Z_3 = abacaba$ encounters $P$. This naturally gives rise to the following definition of an avoidmer.

## Definition (Avoidmer)

A k-mer w is called **$Z_n$-avoidmer** or **$n$-avoidmer** if it avoids Zimin $Z_n$ pattern. In particular, we will refer to a k-mer as **Zimin avoidmer**, or simply *avoidmer* to imply $Z_3$ avoidance.

A natural question to examine, is that given an alphabet $L$ of size $|L| = q$, what is the smallest natural number $f(n,q)$ such that all the words of length $f(n,q)$ encounter the Zimin word $Z_n$. More formally, we define $f(n,q)$ as follows:

$f(n,q) = min \left\{ k \geq 1 : w \text{ encounters } Z_n \text{ for any } w \ \epsilon \ L \text{ such that } |w| \geq k \right\}$.

Again, for our particular case, we will shift our attention to the nucleotide alphabet $q = 4$, and in particular for Zimin patterns $n \leq 3$.

We will list below some known results and properties and asymptotic upper bounds for $f(n,q)$:

- $f(1,q) = 1$

- $f(2,q) = 2q + 1$

- $f(3,q) \leq \sqrt{e} 2^q (q+1)! + 2q + 1$ [25]

In the case of the nucleotide alphabet $q = 4$, we have the following asymptotic bounds,

- $f(1,4) = 1$

- $f(2,4) = 9$

- $f(3,4) \leq 3,174$.

We denote by $f^{\sim}(3)$ the smallest *observed* natural number such that all k-mers that emerge in the human genome of length at least $f^{\sim}(3)$ encounter the Zimin word $Z_3$. By definition,

$$f^{\sim}(3) \leq f(3,4) \leq 3,174.$$

In fact, $f^{\sim}(3)$, in the human genome, is much lower than this upper bound, and unavoidance can be achieved much earlier at 105 bp. In addition, we note that, Zimin pattern *avoidance*, is a symmetric property, in the sense that, a kmer *w* avoids

$Z_n$, if and only if it's reverse complement $r(w)$ avoids $Z_n$. Zimin sequences are symmetric by nature. As such, we have performed the avoidmer analysis on a single strand.

## Results

### Identification of Zimin avoidmers in the human genome

It is established theoretically [16] that any sufficiently large word will entail $Z_3$ patterns in it. Thus, we examined the positional preferences of Zimin avoidmers across the human reference genome. We investigated the Zimin avoidmer distribution using the Telomere-to-Telomere (T2T) complete human genome [26] across all chromosomes for $Z_3$ avoidmers, for k-mer lengths above 50bps. We find that the number of $Z_3$ avoidmers declines precipitously with k-mer length (Fig 1a). For instance, we report 4,651,253 $Z_3$ avoidmers of at least 50 bps length, while at 60 bps and 70 bps we report 675,325 and 70,546 $Z_3$ avoidmers, respectively.

We were interested to study how the distribution of the $Z_2$ and $Z_3$ avoidmer occurrences in the human genome compares to the expected distribution. For a given k-mer length, there are a total of $4^k$ possible k-mers. For the first 14 bp k-mer lengths, we exhaustively generated the subset of avoidmers and calculated the theoretical expected probability of such a k-mer arising (Tables A and B in S2 File). Due to the repetitive nature of the human genome, the corresponding actual probability of an avoidmer emerging in the human T2T genome is lower than the theoretical probability, and this difference between expected and observed amplifies with increasing k-mer length and from $Z_2$ to $Z_3$ (Fig 1b and 1c). We conclude that there are fewer $Z_2$ and $Z_3$ avoidmers in the human genome than expected by a theoretical distribution and their number declines precipitously with increasing k-mer length.

### In the reference human genome, every k-mer contains a Zimin word after 104 base-pairs

As a next step, we wanted to identify the minimum length at which all k-mers cannot avoid $Z_3$ motifs. Specifically, we calculated the minimum length in the human T2T genome, after which all k-mers are not avoiding $Z_3$, $f^{\sim}(3)$ (Definition 1), which was calculated at 105 bp, with a maximum Zimin avoidmer length of 104 bp emerging on chromosome 7 (Fig 1d and Table C in S2 File). Theoretically, for an alphabet of size k the quantity $f(3, q)$ is bounded from above by $\sqrt{e}2^q(q+1)! + 2q + 1$ [19]. Thus, $f(3, 4)$ - and consecutively $f^{\sim}(3)$ - is *at most* 3,174 bp long [19], which is significantly higher than the real $f^{\sim}(3)$ value of 105 bp. This discrepancy can be attributed to the fact that the human genome is a finite sequence that is highly repetitive and has a compositional bias with higher AT than GC content. We find that Zimin avoidmers are significantly more GC rich than the human genome average (Fig 1f). Additionally, when comparing Zimin avoidmers with regions that do not contain any Zimin avoidmer, they are significantly more GC-rich (Fig 1e; Mann-Whitney U, p-value<0.0001).

We also performed genome simulations controlling for the k-mer composition from one up to nine base pairs. We observed that permuting the genome leads to an enrichment of Zimin avoidmer density across all examined k-mer lengths compared to the reference genome. In particular, we discover that Zimin avoidmers have a preference for balanced GC content that does not deviate in either high or low content (Fig A in S1 File). The total Zimin avoidmer density decreases sharply with increasing simulated k-mer length (Fig Ba and Bb in S1 File, one-tailed binomial-test p-value<0.001). Additionally, the maximum observed length of Zimin avoidmers is higher in all permuted genomes, with fixed mononucleotide composition displaying the maximum Zimin avoidmer length of 119 base pairs, accounting for approximately 14% total increase from the human reference genome (Fig Bc in S1 File). Thus, we conclude that the furthest we deviate from the reference genome, achieved using lower order permutations, the higher the Zimin avoidmer density and the longer the maximum observed Zimin avoidmer length. We conclude that every subsequence of the human genome, of at least 105 bps, contains a $Z_3$ Zimin pattern.

### Zimin avoidmer enrichment in specific genomic compartments

Next, we examined if Zimin avoidmers are uniformly distributed in the human genome or if there are specific chromosomes and genomic subcompartments that display an excess. We first investigated if there exist significant differences

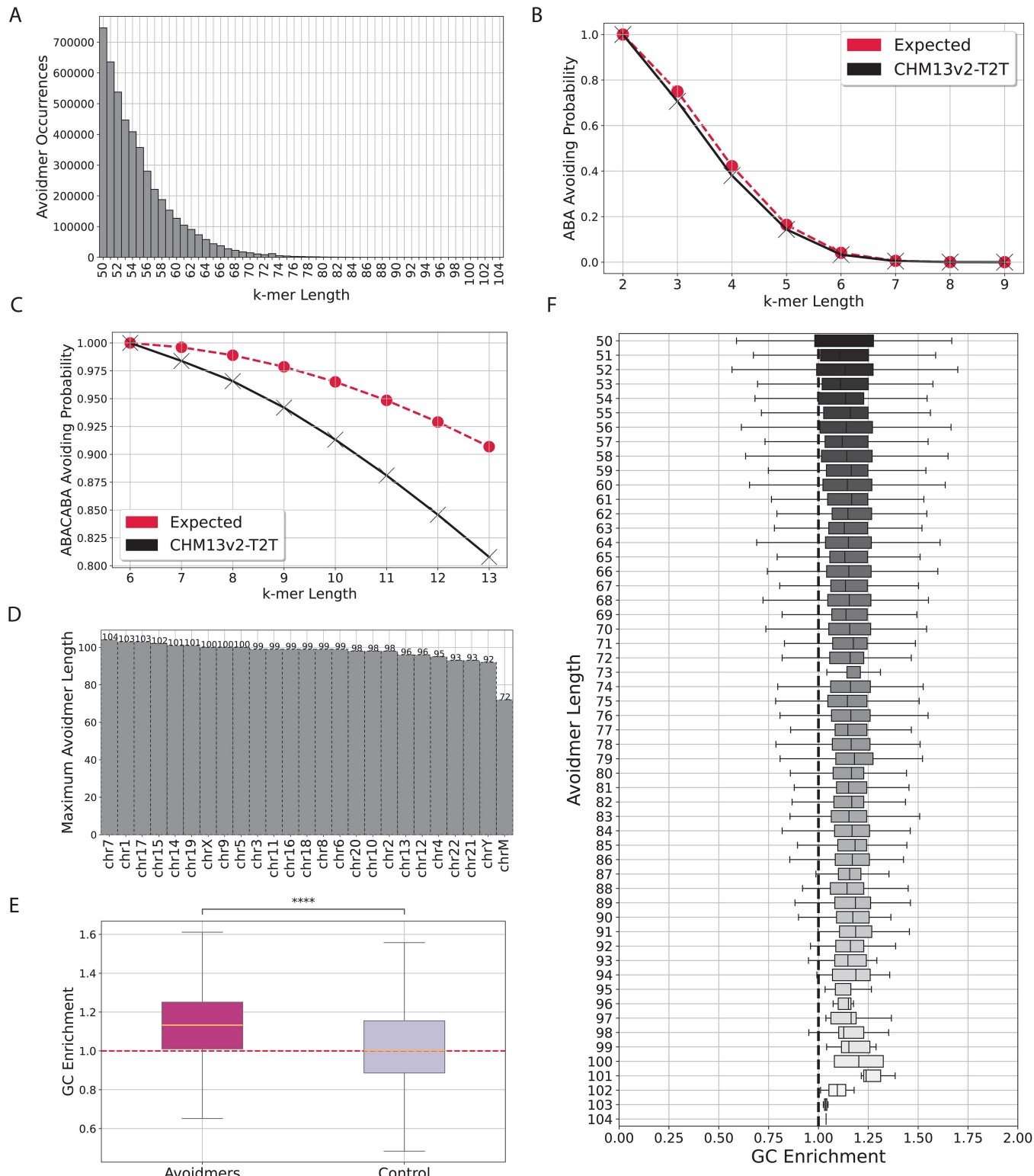

**Fig 1. Identification of Zimin words and Zimin avoidmers in the human genome. A)** Number of Zimin avoiding k-mers as a function of k-mer length, for k-mers ranging between 30 and 100bps. **B-C)** Expected and observed probability of a k-mer being Zimin avoiding for B. n = 2 and C. n = 3. **D)** Highest k-mer length at which avoidmers are observed in each chromosome in the human genome. **E)** Comparison of GC enrichment in regions containing Zimin avoidmers versus regions lacking Zimin avoidmers of at least 50 bp in length. **F)** Enrichment of GC content in avoidmers over the human genome average, for avoidmers of increasing length.

in the distribution of avoidmers among the human chromosomes. We found that Zimin avoidmers of lengths of at least 60 bps have the highest genomic density in the Y chromosome, whereas shorter and longer Zimin avoidmers are more evenly spread between the remaining human chromosomes (Fig 2a).

We also investigated the density of avoidmers in various genomic subcompartments, such as genic, exonic, and coding regions, as well as various pericentromeric and centromeric satellite regions, in order to identify potential hotspots of Zimin avoidmers. The vast majority of Zimin avoidmers of length equal or greater than 50 bps were located in coding sequence (CDS) regions (Fig 2b). When increasing the base pair threshold, Zimin avoidmers of at least 70 bp long, were most enriched in classic human satellite hsat1B covering 2.6% of the satellite regions with an average density of 18,247.77 bp per Mb. The second most enriched genomic region was exonic regions displaying an average density of 3,848.10 bp per Mb (Fig 2b). Despite the fact that satellite compartments are the main source of avoidmer elements, exonic and CDS loci are the third and second, respectively, most enriched genomic subcompartment in avoidmer population. Additionally, in the classical satellite hsat1B, the Zimin avoidmer density shows a sharp drop at 73 bp, whereas the avoidmer distributions in exonic and coding regions exhibit a more gradual decline (Fig 2b and 2c). This could be attributed to the fact, that, due to the repetitive nature of hsat1B the likelihood of $Z_3$ motif eventually emerging increases disproportionally at higher k-mer lengths. The Y chromosome is the richest chromosome in hsat1B regions, and we find that a small subset of Zimin avoidmers originates in these highly repetitive regions (Fig C in S1 File) [27]. Most of the Zimin avoidmer satellite sequences detected in hsat1B are exactly the same sequence of 73 bp long, appearing 4,212 times (Table D in S2 File). Interestingly, the second most frequent Zimin avoidmer in hsat1B is also 73 bp long and has 277 occurrences, and differs from the first most frequent sequence by a single bp substitution (Table D in S2 File). These two sequences explain the sharp drop in Zimin avoidmer density in the classical satellite hsat1B that we observed previously. We conclude that Zimin avoidmers have an inhomogeneous distribution in the human genome, and are most enriched in CDS and hsat1B regions.

## Zimin avoidmers are associated with increased sequence diversity

Next, we binned the genome in non-overlapping 50 kilobase (kB) windows to examine how the k-mer diversity influences the number of avoidmers detected. In each bin we counted the number of unique k-mers detected for six to nine bps k-mer length. Then, we quantified the correlation between the avoidmer occurrences and distinct k-mer sequences detected in each bin. We observe that the Zimin avoidmers with the largest number of occurrences are primarily observed in bins in which there is large k-mer diversity (Figs 3a and D in S1 File). These results are also consistent in CDS compartments (Figs 3b and E in S1 File). However, we find that a small subset of Zimin avoidmers is found in bins that have low k-mer diversity and these originate from satellite repeat regions, such as hsat1B (Figs 3b and E in S1 File). The results were consistent for the different window sizes tests.

Next, to examine potential repetitive patterns in Zimin avoidmers, we investigated the density of subsequences that are an exact instance of $Z_2 = aba$ in Zimin avoidmers [28]. We observe that the density of ABA sequences in Zimin avoidmers follows a normal distribution; however, with an elongated tail representing a small subset of Zimin avoidmers that have high density of ABA sequences (Fig Fa in S1 File). Interestingly, the average ABA density was anti-correlated with the number of distinct k-mers present in each Zimin avoidmer (Fig Fb in S1 File), indicating that, on average, Zimin avoidmers without ABA words have the highest k-mer diversity. In fact, on average, we anticipate the same effect in any k-mer. We conclude that Zimin avoidmers are associated with loci that have increased sequence diversity in the human genome.

We were interested to determine how long, on average, a uniformly randomly generated k-mer would be before it encounters an instance of $Z_3$. We simulated a simple random process which constructs a k-mer by choosing at random - with probability 0.25 - a nucleotide from the alphabet {a, g, c, t}. This random process continues until the resulting sequence encounters an instance of $Z_3$. We repeated this process 50,000 times and documented the resulting Zimin avoidmers for each of the random experiments. We estimated the average value of maximum length of randomly generated Zimin avoidmers being equal to 26.6 bp (Fig Ga in S1 File). However, as quite a few outliers reach higher lengths (Fig

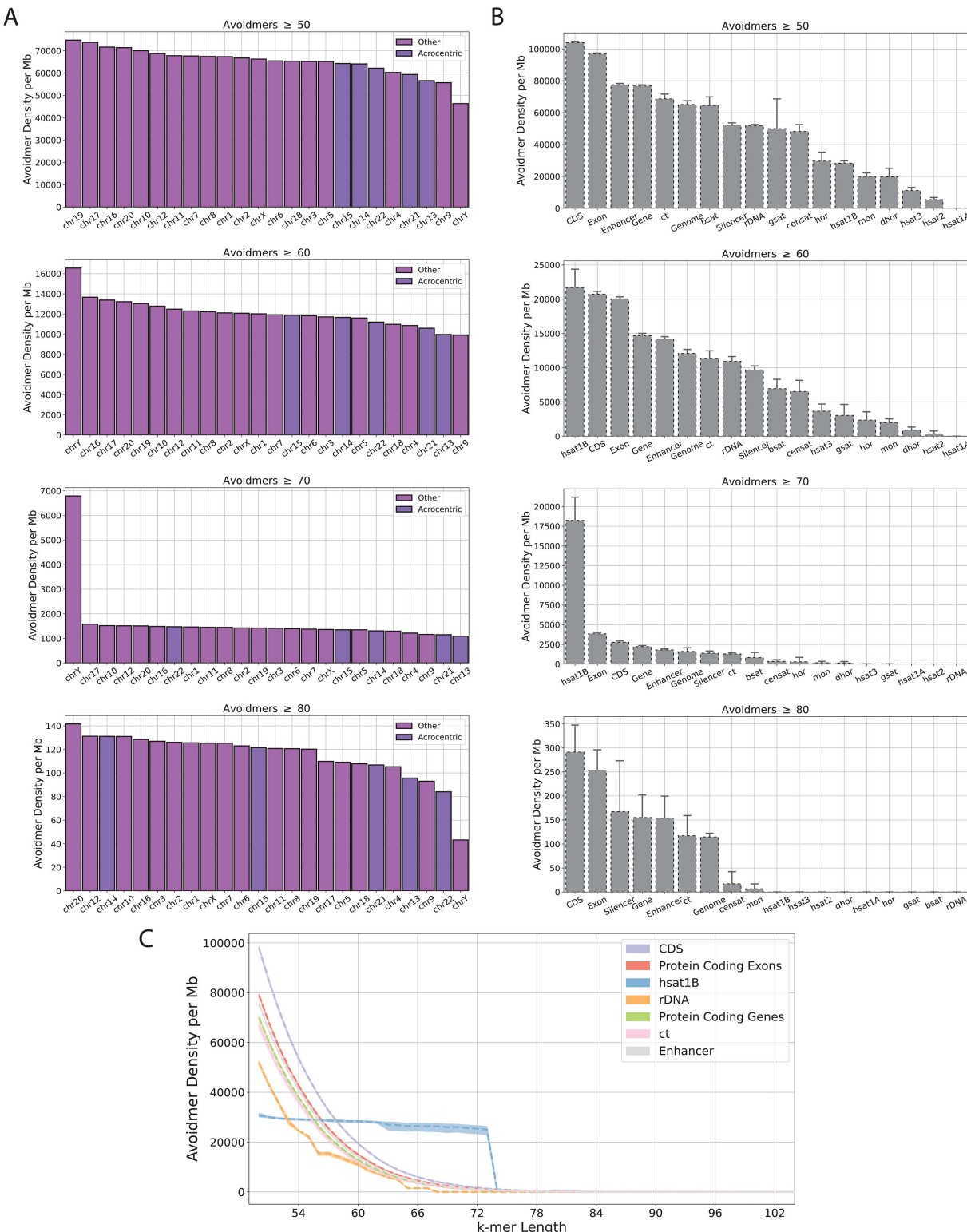

**Fig 2. Zimin avoidmers are inhomogeneously distributed in the human genome. A)** Avoidmer density per Mb across various human T2T chromosomes for k-mer lengths of at least 50, 60, 70 and 80 bp. **B)** Avoidmer density per Mb in human pericentromeric and centromeric satellite and genic subcompartments for k-mer lengths of at least 50, 60, 70 and 80 bp, respectively. **C)** Zimin avoidmer k-mer density as a function of k-mer length across the genomic compartments. For each genomic compartment we used bootstrap N = 250 with replacement to construct confidence intervals for the avoidmer density. Error bars represent 95% confidence interval confidence.

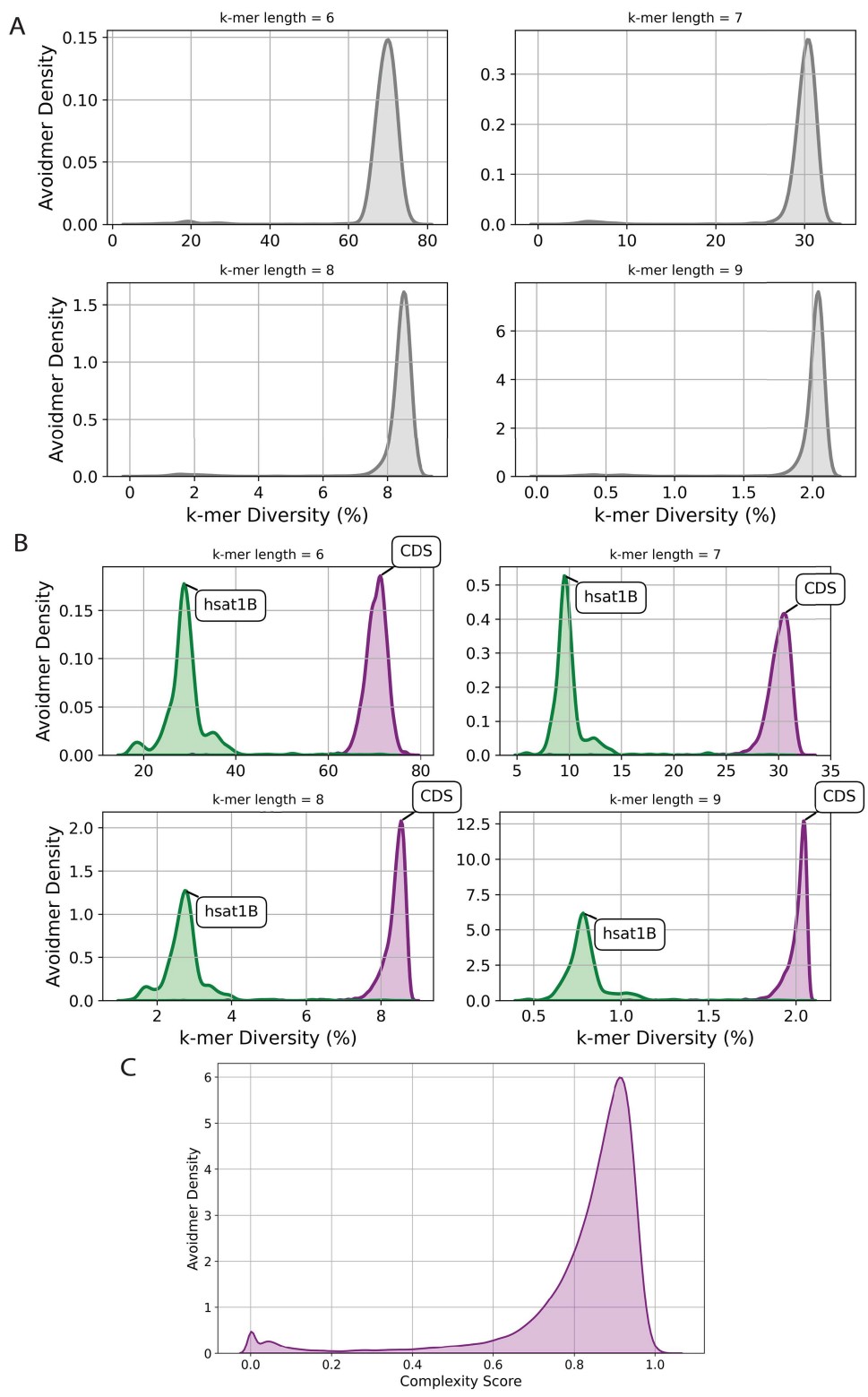

**Fig 3. Characterization of the sequence diversity in Zimin avoidmers. A.** K-mer diversity scores of 50kB regions that contain a Zimin avoidmer of at least 50 bp long, for k-mer lengths between six and nine base-pairs. **B-D.** K-mer diversity scores of 50kB regions that contain a Zimin avoidmer for k-mer lengths between six and nine base-pairs at two compartments: **B.** CDS, hsat1B classic satellite compartments. **C.** Kernel density estimation of avoidmer density distribution, showing the complexity score per 1 kb genomic bins, based on Zimin avoidmers with a minimum length of 50 bp.

Gb in S1 File). Finally, we examined the relationship of the lengths of the resulting sequences with the ABA density. The latter indicated a logarithmic relationship with the maximum lengths, suggesting that a small increase in the total length induces an exponential increase in the subsequences that are instances of $Z_2$ (Fig Gc in S1 File). Although this random process relies on basic assumptions that do not apply to regular genomes, such as nucleotide positional and local independence, it enables us to quantify the inherent difficulty of constructing arbitrarily long Zimin avoiding k-mers by relying purely on chance.

We also hypothesized that due to their inherent complexity, Zimin avoidmer sequences are abundant in parts of the genome with increased complexity. To test this hypothesis, we used data from Pirogov et al. [29], in which the human genome is partitioned into mutually exclusive 1kB bins, each bin assigned a complexity value. We mapped Zimin avoidmer sequences of at least 50 bp long, to a complexity bin. We report that the distribution of complexities of the mapped regions is highly shifted towards the right, meaning that Zimin avoidmers have a tendency to occur in high complexity regions (Fig 3c). These findings support our hypothesis and suggest that Zimin avoidmers preferentially localize to structurally or compositionally complex regions of the genome.

### *In silico* saturation and germline mutagenesis of Zimin avoidmers in the human genome

We investigated if the pattern of being a Zimin avoidmer is resistant to the introduction of single bp indels and substitutions, a property that we hereafter term invariance. We simulated all possible one bp indels and substitutions at each Zimin avoidmer. Zimin avoidmers exhibited a higher rate of invariance for deletions, followed by insertions, and lastly, substitutions (Fig 4a). On average, when increasing the sequence length by randomly inserting a random letter from the nucleotide alphabet the probability of the Zimin property diminishes, due to the fact that it is less likely for a longer sequence to be a Zimin avoidmer (Fig 1a). Thus, it is more likely for deletions to maintain that initial invariance, in contrast to random insertions or substitutions (Fig 4a). Furthermore, due to the fact that for every Zimin avoidmer by definition, each subsequence must also be a Zimin avoidmer, deletions at the end of the k-mer sequence must inevitably maintain the Zimin avoidmer property (Fig H in S1 File).

Subsequently, we used germline variants from The Genome Aggregation Database (gnomAD) [30] and examined the positioning of Zimin avoidmers relative to substitutions and indels. We constructed a 1kB window centered at the germline mutations and counted the number of Zimin avoidmer occurrences in each position. We observe a strong depletion of Zimin avoidmers at the proximal region surrounding insertions and deletions (Fig 4b). Interestingly, we do not find strong enrichment or depletion patterns for substitutions (Fig 4b). Due the complex nature of sequences that avoid $Z_3$ patterns, we postulated that the strong depletion signal of indels originates from absence of Short Tandem Repeats (STRs) at Zimin avoidmers, which are highly enriched for indels [31,32] and depleted in $Z_3$ avoidmers. We examined the relative positioning of STRs in respect to Zimin avoidmer sequences. We performed a comparison of STRs in Zimin avoiding regions and genomic regions that do not contain Zimin avoidmers. We report that STRs are significantly depleted in Zimin avoiding regions (two-tailed Fisher's Exact Test, p-value < 0.001; Fig I in S1 File). Additionally, we found that, as expected, STRs are depleted in the proximity of Zimin avoidmer sequences, particularly in CDS and genic compartments (Fig 4c). This result coincides with our intuition suggesting that due to the inherent complexity of CDS compartments, Zimin avoidmers are most enriched in those regions and are depleted from the repetitive STR loci, impacting their mutation rates.

We conducted an analysis to investigate the association between mutation rate in the human genome and Zimin avoidmer patterns. At first, we used hypermutable sites predicted by Roulette [33] and examined if these loci tend to co-localize within Zimin avoidmer defined genomic regions. We report that out of 2,020 hypermutable sites predicted by Roulette, the 6.25% (n = 161) lie within Zimin avoidmer loci, which constitutes a 1.3-fold enrichment from the genome-wide background rate (p-value < 0.001, Fisher's exact test).

We examined if the Zimin avoidmer regions exhibit higher mutation rates than expected, we constructed two control sets: randomized shuffled controls (Random group), and GC-matched controls (GC group). We conducted an

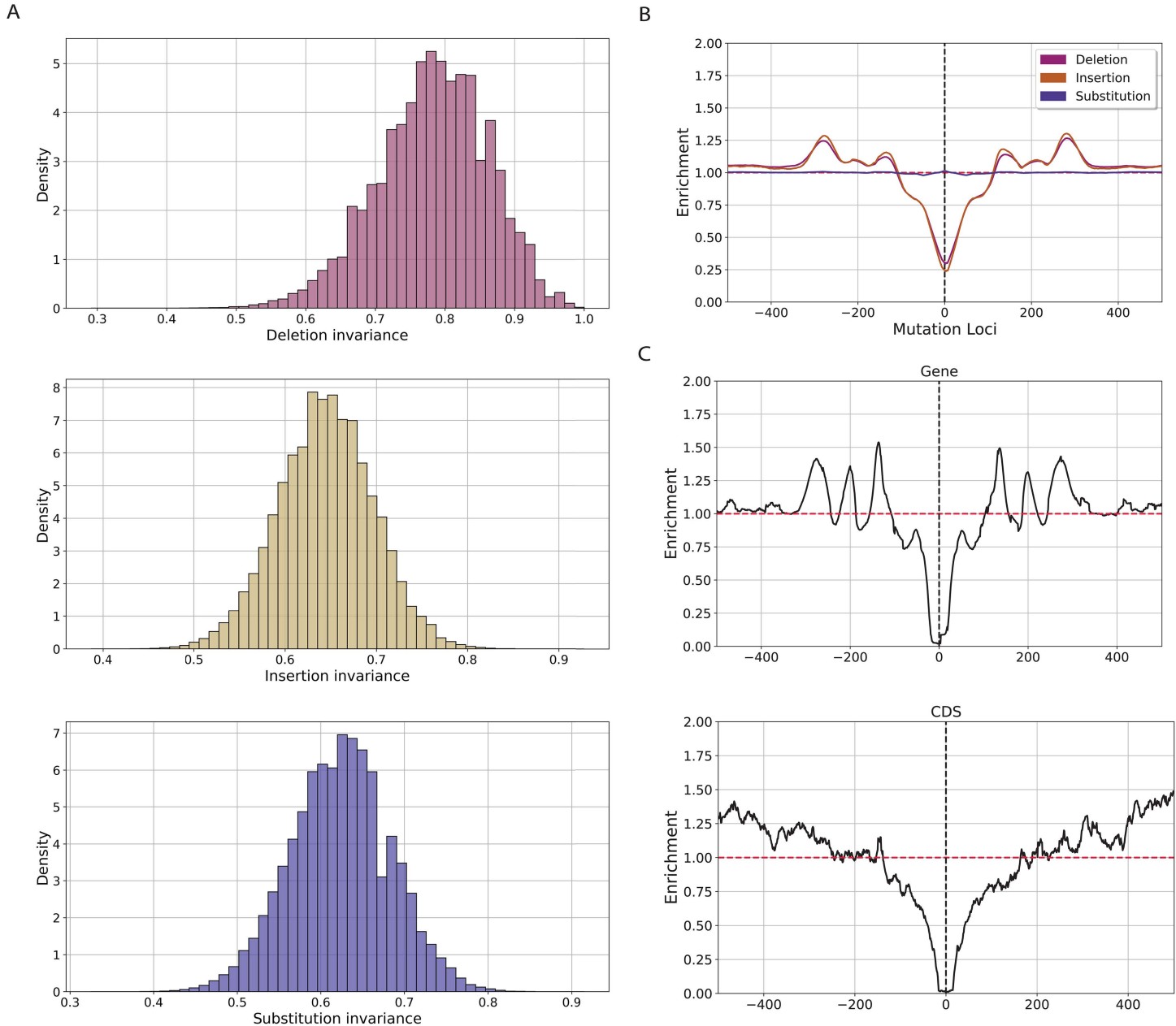

**Fig 4. A) Histogram of invariance of avoidmers under mutational transformations. B)** Relative positioning of avoidmers in relationship to substitutions, deletions and insertions across 1kB window. **C)** Relative positioning of STRs in relationship to genic and CDS avoidmers, respectively.

investigation of the distribution of predicted mutation rate by Roulette density within Zimin avoidmers and each afore-mentioned control group. We report that all groups exhibit a bimodal distribution with Zimin avoidmers, followed by the GC-matched control group exhibiting a slightly higher concentration of higher mutation rate (Fig Ja in S1 File). Additionally, we investigated if the mutation rates differ across the genomic subcompartments. We report that, while Zimin avoidmers exhibit significantly higher mutation rates than the two control groups across genic, exonic and coding regions (two-tailed Mann-Whitney U, adjusted for multiple comparisons using Benjamini-Hochberg procedure) (Fig Jb in S1 File), the

common language effect size differences between the Zimin avoidmers and the two constructed control groups suggest that the observed differences are negligible (Table E in S2 File).

### Identification of Zimin words and Zimin avoidmers across model organism genomes

Next, we examined how our findings translate in other organismal genomes. We selected a set of reference genomes from model organisms, spanning all three domains of life (Table 1), and estimated the k-mer length after which every k-mer contains a Zimin word. We observe that the highest Zimin avoidmer length is observed for *S. cerevisiae* for k-mer length of 115 bps, whereas the lowest is for *S. aureus* at 86 bps (Fig 5a and Table 1). We also find that the $Z_3$ avoidmer density of Zimin avoidmers of at least 50 bp length, varies substantially between the examined organisms, ranging between 180,905.07 and 58,493.34 $Z_3$ avoidmers per mB in *E. coli* and *D. rerio* respectively (Fig 5b). We investigated the expected theoretical probability and observed probability of observing Zimin avoidmers in each of the model organisms. We find that the largest discrepancies between expected and observed probabilities are in eukaryotic organisms including *G. galus*, *D. rerio*, *D. melanogaster*, *S. cerevisiae* and *C. elegans*, whereas the smallest differences are observed in prokaryotes, namely *S. aureus*, *K. pneumoniae* and *E.coli* (Fig 5c and 5d and Table F in S2 File). We shuffled each genome while controlling for nucleotide composition across k-mer lengths ranging from one to seven nucleotides. We observed that genomes shuffled with fewer constraints, specifically, those controlling for shorter k-mer lengths (lower k), converged more closely to the theoretical probability expected for Zimin avoidmers (Fig 5c and Table F in S2 File, paired sample t-test, adjusted p-value < 0.05). This can be likely explained by the more repetitive and nucleotide imbalanced genomes of eukaryotes compared to prokaryotes.

### Zimin avoidmers are inhomogeneously distributed across model organismal genomes

For each examined organism we separated their genome in genic and intergenic regions and examined the percentage of Zimin avoidmers in each normalizing for region length. We find that across organisms, Zimin avoidmers have more occurrences in genic than intergenic regions (Fig 5e and 5f), results that are consistent with our observations for the human genome. When further separating the genomic compartments in genic, exonic and cds regions, we find that the Zimin avoidmer density varies substantially both between sub-compartments and between the species (Fig 5g). Additionally, we report that Zimin avoidmer density increases with the CDS to genome ratio of an organism (Fig K in S1 File; Pearson correlation: r = 0.72, p-value < 0.05). We conclude that across most of the studied organisms, Zimin avoidmers are most enriched in genic and CDS regions of the genome, particularly in the case of organisms of eukaryotic origin.

## Discussion

We have characterized for the first time Zimin words and Zimin avoidmers in the genomes of nine organisms, including the human genome and the genomes of multiple model organisms. This is a mathematical concept, with significant previous theoretical research [17–19], that is applied for the first time in a genomics setting. By construction, genomic sequences that avoid Zimin patterns capture a higher level of grammar complexity, since they are bound to avoid certain repetitive patterns in order to maintain their Zimin avoidability property. Furthermore, on sufficiently large genomic sequences, the unavoidability theorem suggests that the emergence of such patterns is unavoidable [16]. We observe that Zimin avoidmers disappear from organismal genomes at lower k-mer lengths than expected from the theoretical upper limit (Table 1). This is likely driven by the repetitive nature of organismal genomes and the uneven frequency of nucleotides. We also observe that Zimin avoidmers are inhomogeneously distributed in organismal genomes. In the human genome, Zimin avoidmers are over-represented in the Human Satellite 1B compartments and in coding sequences.

Zimin avoidmer sequences, by construction, do not contain any STR as a proper subsequence. Thus, these artificial nucleotide regions encapture a higher level of irregularity which is interrupted by canonical patterns of DNA, such as

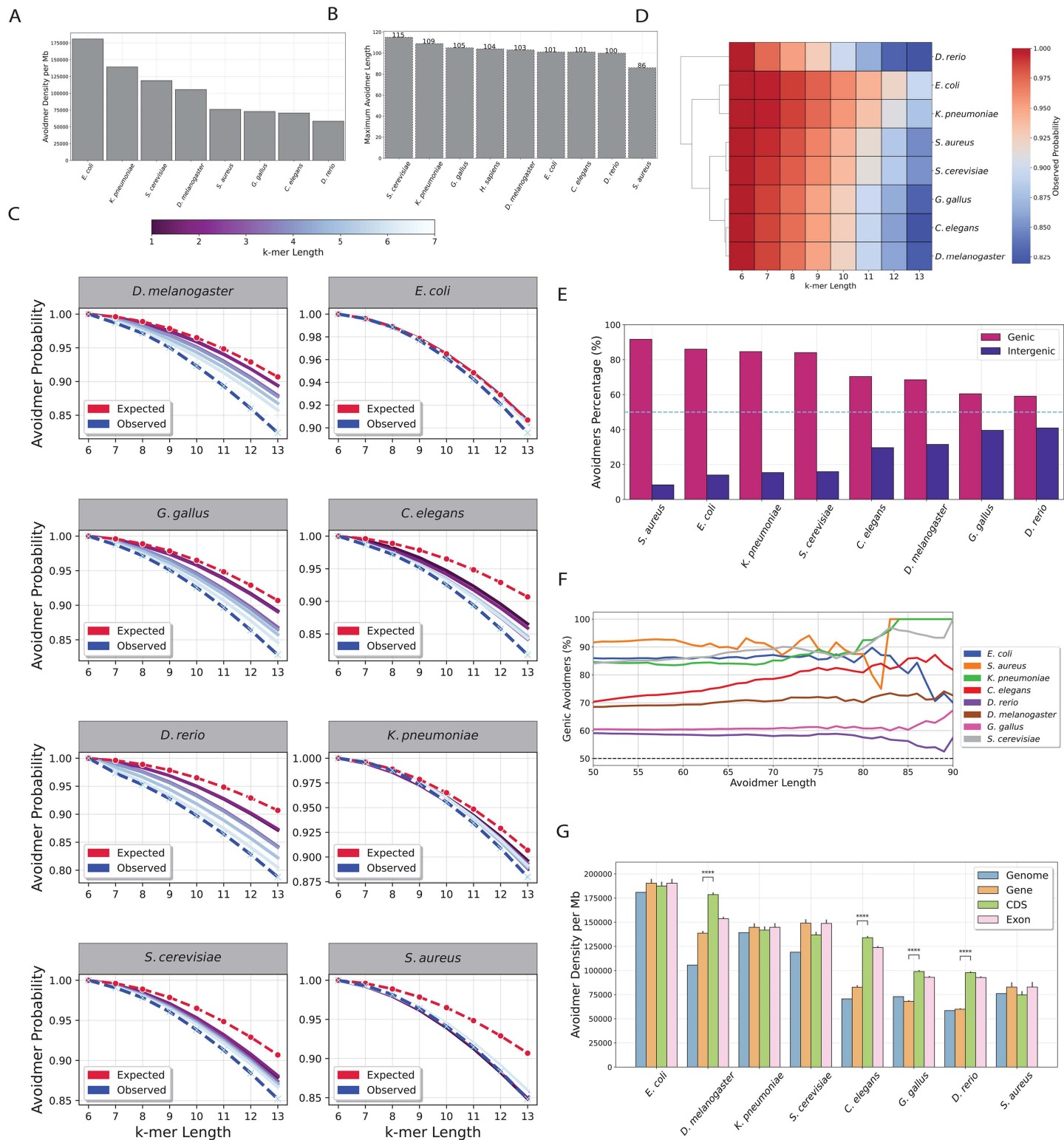

**Fig 5. Identification of avoidmers across model organisms. A)** Highest k-mer length at which avoidmers are observed in each chromosome in the human genome. **B)** Density of Zimin avoiding k-mers across model organisms. **C)** Expected and observed probability of k-mer being Zimin avoiding across model organisms. Colored lines represent the simulated genomes controlling but different k-mer lengths between one and seven base pairs. **D)**

Observed probability of a k-mer being avoidmer across model organisms. **E)** Avoidmer percentage across genic and intergenic regions for each model organism. **F)** Zimin avoidmer percentage for increasing k-mer length for each of the under study model organisms. **G)** Zimin avoidmer density across genomic subcompartments for each model organism. Zimin avoidmer density was compared between genic and coding regions using independent t-tests, with multiple testing correction applied via the Benjamini–Hochberg procedure.

**Table 1. Highest k-mer length at which avoidmers are observed in each model organism's genome.**

| Species | Avoidmer threshold (bp) |
| --- | --- |
| *Saccharomyces cerevisiae* | 115 |
| *Klebsiella pneumoniae* | 109 |
| *Gallus gallus* | 105 |
| Homo sapiens | 104 |
| *Drosophila melanogaster* | 103 |
| *Caenorhabditis elegans* | 101 |
| *Escherichia coli O157:H7* | 101 |
| *Danio rerio* | 100 |
| *Staphylococcus aureus* | 86 |

tandem repeats. The total density of Zimin avoidmers is more pronounced in prokaryotic genomes rather than eukaryotic genomes, which could potentially be related to the depletion of microsatellite sequences in certain bacterial phyla [34].

Zimin sequences and their counterparts, the Zimin avoidmers, are of particular interest and hold potential utility in bioinformatics. Their inherent symmetry and self-similarity could link them with non-canonical DNA conformations, also known as non-B DNA motifs (Wang and Vasquez 2023), as these patterns exhibit both direct and mirror-repeat symmetries. This raises the question of whether the presence or absence of such properties in k-mers plays a crucial role in the formation of various DNA secondary structures across different genomes. We believe that the characterization of Zimin words provides a new lens through which one can view genomes using homomorphism embeddings of various patterns.

The ability of Zimin avoidmers to exclude repetitive patterns presents intriguing opportunities for synthetic biology and genome engineering. Synthetic DNA constructs often require the design of sequences with high sequence complexity, low repeat content, and resistance to mutations, to ensure stability and minimize off-target effects [22,35,36]. Given that Zimin avoidmers are complex sequences, and are depleted of repeats and indel mutations, Zimin avoidmers could be used to design mutation-resistant, low-repeat-content synthetic sequences, offering potential advantages for applications in biotechnology, synthetic biology, and genetic engineering. Additionally, their inherent resistance to insertions and deletions makes them attractive candidates for molecular barcoding, synthetic promoters, or as neutral scaffolds in genome editing platforms. In genome studies, Zimin words may serve as a theoretical tool to explore genomic complexity and repeat structure, and provide insights into the principles of genome architecture and evolution.

Future studies could implement additional concepts from algebraic combinatorics on words in genomics to acquire a deeper understanding of the rules of genomic grammar. Finally, future work is required to further examine Zimin words in organismal genomes and their emergence or loss through population variants, as well as potential applications and development of tools based on them.

## Methods

### Data retrieval

We downloaded the reference human genome assembly T2T-CHM13v2.0. Associated files including gene annotation and comprehensive centromere/satellite repeat annotation files were downloaded from https://github.com/marbl/

CHM13. The gene annotation GFF file was downloaded from https://ftp.ncbi.nlm.nih.gov/genomes/all/GCF/009/914/755/GCF_009914755.1_T2T-CHM13v2.0/GCF_009914755.1_T2T-CHM13v2.0_genomic.gff.gz and a more comprehensive centromere/satellite repeat annotation file was derived from https://s3-us-west-2.amazonaws.com/human-pangenomics/T2T/CHM13/assemblies/annotation/chm13v2.0_censat_v2.0.bed. We also downloaded the following complete genomes: *Haloferax volcanii DS2* from https://ftp.ncbi.nlm.nih.gov/genomes/all/GCF/000/025/685/GCF_000025685.1_ASM2568v1/GCF_000025685.1_ASM2568v1_genomic.fna.gz, *Halobacterium salinarum* from https://ftp.ncbi.nlm.nih.gov/genomes/all/GCF/004/799/605/GCF_004799605.1_ASM479960v1/GCF_004799605.1_ASM479960v1_genomic.fna.gz, *C. elegans* from https://hgdownload.soe.ucsc.edu/goldenPath/ce11/bigZips/ce11.fa.gz, *D. melanogaster* from https://hgdownload.soe.ucsc.edu/goldenPath/dm6/bigZips/dm6.fa.gz, *G. gallus* from https://hgdownload.soe.ucsc.edu/goldenPath/galGal6/bigZips/galGal6.fa.gz, *S. cerevisiae* from https://hgdownload.soe.ucsc.edu/goldenPath/sacCer3/bigZips/sacCer3.fa.gz, Zebrafish from https://hgdownload.soe.ucsc.edu/goldenPath/danRer11/bigZips/danRer11.fa.gz, *Klebsiella pneumoniae* subsp. pneumoniae HS11286 from https://ftp.ncbi.nlm.nih.gov/genomes/all/GCF/000/240/185/GCF_000240185.1_ASM24018v2/GCF_000240185.1_ASM24018v2_genomic.fna.gz, *Escherichia coli* O157:H7 from https://ftp.ncbi.nlm.nih.gov/genomes/all/GCF/000/008/865/GCF_000008865.2_ASM886v2/GCF_000008865.2_ASM886v2_genomic.fna.gz, *Staphylococcus aureus* subsp. aureus NCTC 8325 from https://ftp.ncbi.nlm.nih.gov/genomes/all/GCF/000/013/425/GCF_000013425.1_ASM1342v1/GCF_000013425.1_ASM1342v1_genomic.fna.gz. We also downloaded GTF files for gene annotation for each of these organismal genomes. Single nucleotide substitutions, insertions and deletions were downloaded for the Human Pangenome Reference Consortium from year one data from https://s3-us-west-2.amazonaws.com/human-pangenomics/pangenomes/freeze/freeze1/minigraph-cactus/hprc-v1.1-mc-chm13/hprc-v1.1-mc-chm13.vcfbub.a100k.wave.vcf.gz.

## Identification of Zimin avoiding sequences

To extract the *Zimin avoiding sequences* across the different kmer lengths, we recursively constructed a backreferencing regular expression pattern matching using Python Table 2. For each chromosome, we scan the sequence from left to right, reading chunks of k-mers of at least 50 bp long. For each of these chunks, the Zimin avoidance property is examined. We are attempting an extension by reading one more base pair at a time, and testing Zimin avoidance. We continue this extension process until one additional bp breaks Zimin avoidance. Before we exit the loop, we save the k-mer without the final base pair, as Zimin avoidmer. The process continues until we have exhausted the whole genomic sequence. Overlapping sequences are merged using pybedtools [37] before analyzing and reporting the genomic sub-compartment densities for the studied organisms. The GitHub link to the code that performs the detection of Zimin avoidmers is provided in the relevant section).

## Estimation of expected and observed Zimin word frequencies

The probability of observing a Zimin word within a particular organismal genome G was estimated using the total number of avoidmers for a particular k-mer length divided by the total number of k-mer occurrences for that particular genome, i.e.,

**Table 2. Regular expression pattern used to extract Zimin avoiding sequences.**

| Length | Regular expression pattern | Zimin avoidmer |
|---|---|---|
| n = 1 | ([agct]+) | $Z_1 = a$ |
| n = 2 | ([agct]+)([agct]+)\1 | $Z_2 = aba$ |
| n = 3 | ([agct]+)([agct]+)\1([agct]+)\1\2\1 | $Z_3 = abacaba$ |
| n = 4 | ([agct]+)([agct]+)\1([agct]+)\1\2\1([agct]+)\1\2\1\3\1\2\1 | $Z_4 = abacabadabacaba$ |

$$P_G(Z_3) = \frac{total\ avoidmers}{total\ number\ of\ k-mers}.$$

The k-mer occurrences were calculated using Jellyfish [38] for k-mer length between 1 and 13 bp for each organismal genome in this study.

The expected probability was calculated by recursively generating the nucleotide k-mer tree while pruning sequences that are not Zimin avoidmers. At each recursive step, the total number of k-mers that belong to the avoidmer class were calculated and subsequently divided by the total number of k-mers for that particular genome to derive the expected probability.

### Examination of avoidmers in genomic compartments

To assess the genomic context of Zimin avoidmers, we identified their coordinates in the genome and quantified overlaps with various genomic subcompartments, requiring a minimum of one base-pair overlap between them. The density of Zimin avoidmers in genic, exonic, 5' UTR, 3' UTR, and CDS regions as well as in centromeric and pericentromeric repeat regions was calculated using publicly available GTF and GFF coordinate files. Additionally, for the prokaryotic genomes we used agat [39] to fill up the missing exon annotations. For each annotated subcompartment, overlapping compartments of the same type were merged and expanded into non-overlapping, mutually disjoint sets of coordinates. These coordinates were then used to estimate the density per Mb by calculating the total number of bps of avoidmers divided by the total compartment length of the previously merged genomic regions.

### Quantification of mutation frequency in the vicinity of Zimin words

We generated a 500 bp window upstream and downstream from each Zimin word's genomic coordinates center and estimated the mutation rate at each bp at each bp. The enrichment was calculated as the number of mutation occurrences at a position over the mean number of occurrences across the window.

### Controls for GC enrichment comparison

We defined genomic regions in the human genome that do not contain any Zimin avoidmer sequence of at least 50 bp long, using the BEDTools subtract command [40]. We compared the GC-enrichment of these genomic regions to that of Zimin avoidmers of at least 50 bp long, and used Mann-Whitney test to evaluate the statistical significance (Fig 1e; Mann-Whitney U, p-value<0.0001).

We partitioned the genome into mutually exclusive regions of 2,000 kB. For each region, we calculated the GC proportion and the Zimin avoidmer fold enrichment by dividing the region's observed Zimin avoidmer density by the genome-wide density. We used a second-degree polynomial model to capture the relationship between GC content and Zimin avoidmer fold enrichment, employing the scikit-learn Python library. To determine the optimal region bin size and polynomial degree, we conducted a grid search with 10-fold cross-validation, using a polynomial feature transformer combined with a linear model.

### Quantification of k-mer diversity across the human genome

A k-mer w is defined as any sequence of k letters drawn from the nucleotide alphabet $A = \{a,\ g,\ c,\ t\}$. The length of the k-mer w is denoted as $|w| = k$. Given a k-mer w, we define the subsequence $w[i : i + l]$ of w, as the $l$-mer that starts at position $0 \leq i \leq |w| - l$ and ends at position $i + l - 1$ (inclusive) with $1 \leq l \leq |w|$. Given a k-mer $w$, we define the canonical k-mer $C(w)$, as the k-mer or its reverse-complement that is minimum in respect to the lexicographic order. For a given genomic region $E$ and an integer number $k$, we define the $k$-diversity of $E$, denoted by $div(E, k)$, as the ratio of the number of unique canonical k-mers within $E$ to the total number of possible canonical k-mers $N(k)$. Thus,

$$div(E, \ k) = \frac{\left| \{ w[i:i+k]: \ 0 \le i \le |E| - k + 1 \} \right|}{N(k)},$$

where $N(k) = \frac{4^k - 4^{0.5k}}{2} + 4^{0.5k}$, $k \in \mathbb{N}$, is the total number of unique canonical k-mers.

To investigate the relationship of avoidmers to high complexity genomic areas, we partitioned each chromosome of the *Homo sapiens* T2T assembly into non-overlapping, consecutive 5kB, 10kB, 25kB, 50kB, 100kB window bins, respectively. For each of the bins, we evaluated the associated k-mer diversity as defined above for k = 6, 7, 8, and 9 base pairs. Subsequently, we calculated the number of Zimin avoidmers in each genomic bin. Then we compared the association between the k-mer diversity and the total number of occurrences of Zimin avoidmers.

## Performance of genome simulations

We used the ushuffle package to perform genome simulations while controlling for nucleotide composition based on the frequency of the k-mer content [41]. For each studied organismal genome we generated nine simulated genomes, each time controlling for one up to nine base pair k-mer frequencies. From the resulting simulated genomes, we extracted the Zimin avoidmers of length at least 50 bp long to estimate their genome-wide frequency. Compared to every simulated genome with a fixed k-mer composition, the fold enrichment was calculated as the ratio between the total density of Zimin avoidmer occurrences in the original (unshuffled) genome and their corresponding density in the simulated genome, i.e.,

$$FE = \frac{\#ZA(k)}{\#ZA},$$

where with *#ZA* we denote the total number of Zimin Avoidmer base pairs in the reference genome, and with *#ZA(k)* the total number of Zimin Avoidmer base pairs in the simulated genome with fixed k-mer length composition.

For each genome, the Zimin avoidmer density was estimated by calculating the total number of Zimin avoidmer base pairs divided by the total genome size. To ensure that we do not overcount the same base pairs twice, we merged the overlapping Zimin avoidmers, using BEDTools merge command [40]. Additionally, to evaluate the significance of the observed enrichment between the permuted and the reference genomes, we used one-tailed binomial test, as provided from python SciPy package, using as base probability, the Zimin avoidmer density of the reference genome, the genome size as the sample space size, and the observed Zimin avoidmer base pairs in the simulated genome as the total number of trials.

## Analysis of high complexity regions

To assess if Zimin avoidmers are abundant in high complexity regions, we used data provided from Pirogov et al [29]. We used the provided bigWig file in which the GRCh38 is partitioned into mutually exclusive regions of 1kB long, each region assigned with a complexity score. We used liftOver to map the coordinates GRCh38 to CHM13v2-T2T genome and subsequently utilizing the command "bedtools intersect -wo f=1.0" we mapped the Zimin avoidmers to a unique complexity region. Finally, we plotted the distribution of the complexity scores using kdeplot as provided from Seaborn graphics library.

## Enrichment of STRs in regions at Zimin avoidmers

We used the Zimin avoidmer coordinates to partition the human genome into two mutually disjoint groups: genomic regions that contain Zimin words and genomic regions that avoid them. Additionally, we used non-b GFA [42] to extract STRs in the CHM13v2-T2T human reference genome. We estimated the genome-wide STR density by dividing the total number of STR base pairs by the genome size. To avoid overcounting due to overlapping regions, we first merged

overlapping STR intervals using the bedtools merge command. Subsequently, we used bedtools coverage to estimate the total density of regions containing Zimin words and those that avoid them. For each group, STR density was calculated by summing the number of base pairs overlapping with STRs and dividing by the total number of base pairs in the respective group. Fold enrichment was then computed as the ratio of the observed STR density within each group to the genome-wide STR density. Statistical significance was assessed using a two-sided Fisher's exact test.

## Monte-Carlo simulations to characterize Zimin avoidmer properties

We simulated a random process by which we generated random Zimin avoidmer sequences. We constructed a Zimin avoidmer by repeatedly sampling one character from the nucleotide alphabet $A = \{a, g, c, t\}$ at random. At each step, the resulting sequence was evaluated to satisfy the Zimin avoiding criteria. If at any step, the new nucleotide was violating the Zimin avoiding property, we saved the length of the previous Zimin avoiding sequence. The process was repeated 50,000 times. The resulting sample data were stored in a pandas dataframe and were processed to generate the simulation figures. Additionally, by repeatedly resampling from the simulated dataset with increasing sample size, initiating at 50 and reaching 5,000, with 10 sample increments between iterations, by the Law of Large numbers, we were able to asymptotically evaluate the average length of avoidmer sequences generated by this simple stochastic process. However, note that the generated average length, is of course, bounded to this simple stochastic experiment, and cannot be generalized to sequences not generated by this exact random process. Finally, the ABA subsequence density was evaluated for each of the generated Zimin avoidmer sequences, revealing a logarithmic relationship between the length of the avoidmer sequence and the subsequence ABA density.

## *In silico* saturation mutagenesis in Zimin avoidmers

To determine the mutational invariance of each Zimin avoidmer, we used a custom Python script that exhaustively generates all possible substitutions, deletions, and insertions and examined if the avoidmer property of the resulting k-mer remained invariant. For each Zimin avoidmer we saved the proportion of possible variants that preserved the Zimin avoiding property to the total number of possible variants, separately for insertions, substitutions, and deletions. Additionally, for each avoidmer we examined the proportion of mutations for which the resulting mutated sequence remained an avoidmer, which we termed invariance. Because each Zimin avoidmer has varying length, for each sequence, we partitioned the k-mer length into 22 mutually exclusive bins, and for each bin, the average invariance was evaluated as the average number of mutations that kept the Zimin property.

## Examination of germline variants at Zimin avoidmers

We used the germline mutations from The Genome Aggregation Database (gnomAD) [30]. Each mutation was classified into one of the following categories: Substitution, Insertion, Deletion and MNP. The allele frequency was not taken into account. Subsequently, a 1kB window was constructed, centered at each of the classified mutational loci. Utilizing the pybedtools package and the intersect function, for each of the previously constructed intervals, we used the BED coordinates of Zimin avoidmers, and mapped each of them at each window, whenever an intersection occurred. Finally, we calculated for each position across the 1kB window the number of Zimin avoidmer bps relative to the mutational loci. Finally, we plotted the enrichment of Zimin avoidmers across the 1kB window using the matplotlib and Seaborn Python libraries [43].

## Estimation of predicted mutation rates within Zimin avoidmers and matched controls

For each Zimin avoidmer we created matched randomized controls using BEDTools shuffle command. To assess the influence of GC-bias of Zimin avoidmers, we constructed an additional GC-content matched control group using an

in-house python script. We used predicted mutation rates predicted by Roulette for each chromosome using data from [33]. We downloaded hypermutable sites from http://genetics.bwh.harvard.edu/downloads/Vova/Roulette/hypermutable/. Additionally, we downloaded all the predicted mutation rates from: http://genetics.bwh.harvard.edu/downloads/Vova/Roulette/. Each file was processed and the predicted mutation rates across the mutation spectrum were averaged into a single mutation rate per loci. Subsequently, for each element across the three groups: Zimin avoidmers, randomized controls, GC-matched controls, we calculate a single average mutation rate, using the bedtools map function. The statistical comparisons were conducted using two-tailed Mann-Whitney test from scipy Python package. The visualizations were performed using seaborn Python library. The jitter plot was randomly sampled from each group corresponding to 1% of the total population.

### Zimin avoidmer distribution relative to short tandem repeats

To determine the frequency of Zimin avoidmers relative to STRs, we extracted STRs for human T2T genome chm13v2 as described in [34]. Using the aforementioned procedure, we constructed 1kB intervals centered around the Zimin avoidmers, and examined the number of occurrences of STR bps across the 1kB window, across all Zimin avoidmer loci. We calculated the total occurrences at each position resulting and divided by the mean occurrences across the window to evaluate the enrichment of STR at each position relative to Zimin avoidmers.

### ABA density

The ABA density of a k-mer w, is defined as the ratio of the total number of subsequences of w that are an instance of ABA $Z_2$ motif to the total number of subsequences of w [28], i.e.,

$$\lambda(w) = 2\frac{\#\{subsequences\ that\ are\ instance\ of\ Z_2\}}{N(N+1)},$$

where $N = |w|$. This formula was implemented in a custom C++ script to extract the ABA density for each of the extracted Zimin avoidmer sequences. Note that the function $\lambda(w)$ is bounded above by 1. In fact, the upper bound can be improved. By using the observation that the N mononucleotides and N-1 dinucleotide subsequences cannot possibly form an instance $Z_2$. Thus, by subtracting these $2N-1$ mononucleotide and dinucleotide subsequences, we conclude that:

$$\lambda(w) \leq 1 - 2\frac{2N-1}{N(N+1)}.$$

### Zimin avoiding patterns do not encounter tandem repeats

A tandem repeat is any k-mer which can be written in the form $x^m = x...x$ m times. The x is often referred to as the *consensus sequence* of tandem repeat and number m as the consensus repeats. A microsatellite is defined as any tandem repeat with $|x| \leq 9$. We claim that any tandem repeat of at least 4 consensus repeats encounters a $Z_3$ pattern when the consensus sequence is at least 2 bp long.

For mononucleotides with consensus motif x with $|x| = 1$ and $m \geq 7$, $x^m$ encounters the pattern $Z_3$ for $f(a) = f(b) = f(c) = x$, as $f(Z_3) = f(abacaba) = x^7$ is a subsequence of $x^m$.

For $|x| \geq 2$ and $m \geq 4$, we can expand x in the form $x = yz$, and, consequently $x^m = (yz)^m$. By defining the homomorphism f with $f(a) = y$, $f(b) = f(c) = z$, we note that:

$$f(Z_3) = f(abacaba)$$

$$= f(a)f(b)f(a)f(c)f(a)f(b)f(a)$$

$$= yzyzyzy$$

$$= (yz)^3 y.$$

But $f(Z_3) = (yz)^3 y$ is a subsequence of $x^m$, as

$$x^m = (yz)^m = (yz)^3 yz(yz)^{m-4} = f(Z_3)z(yz)^{m-4}.$$

Thus $x^m$ encounters $Z_3$ for all cases where $|x| = 1$ & $m \geq 7$ and $|x| \geq 2$ & $m \geq 4$. We conclude that avoidmers cannot coincide with the majority of tandem repeats loci.

## Supporting information

**S1 File. Supplementary figures.** Fig A: Relationship of GC-Content and Zimin avoidmer density across the simulated genomes of human telomere-to-telomere reference genome. For each simulated genome, the genome-wide avoidmer density was evaluated, and a second degree linear regression was used to model the relationship between the GC-content and avoidmer density. Fig B: Fold enrichment of avoidmers in simulated genomes of the human telomere-to-telomere reference genome. A) Fold enrichment of avoidmer density of simulated shuffled genomes with respect to the reference genome, displayed in increasing order of preserved nucleotide composition starting from mononucleotides up to nine-nucleotides. The significance of the enrichment was evaluated using one-tailed binomial test, with background base rate being chosen as the the probability of a single base pair belonging to an avoidmer sequence of at least 50 bp long, the number of positive events equal to the number of avoidmer base pairs in the shuffled genome, and the genome size as the total number of iterations. B) Fold enrichment of avoidmer density of simulated shuffled genomes with respect to the human reference genome, as a function of the preserved nucleotide composition, for various minimum avoidmer thresholds. C) Maximum avoidmer length increase across shuffled genomes. Fig C: Avoidmer density across the Y chromosome. The highlighted regions correspond to the classical human satellite region hsat1B where the vast majority of avoidmers are located. The densities of Zimin avoidmer sequences of at least 70 bp long, appear denser in the hsat1B compartments rather than the hsat3 satellite regions. Fig D: ABA sub-sequence density in avoidmers and its relationship with canonical k-mer diversity. A. ABA sub-sequence density in avoidmer sequences. B. ABA density in avoidmers as a function of average canonical k-mer diversity for various k-mer lengths. Fig E: Kernel density estimation of k-mer diversity scores of 5kB, 10kB, 25kB, 100kB regions that contain a Zimin avoidmer of at least 50 bp long, for k-mer lengths between six and nine base-pairs. Fig F: Kernel density estimation of k-mer diversity scores of 5kB, 10kB, 25kB, 100kB regions that belong either to hsat1B (green) or CDS (purple), contain a Zimin avoidmer of at least 50 bp long, for k-mer lengths between six and nine base-pairs. Fig G: Examination of Zimin avoidmer length and composition distribution. A) Average avoidmer length threshold approximation by increasing sample size, following the law of large numbers. Random sampling with replacement was used, with sample sizes ranging between 50 and 5,000, with 10 sample increments between iterations. B) Avoidmer length threshold distribution from random equidistributed simulations using the nucleotide alphabet. C) Average ABA Density is exponentially increased with Zimin avoidmer length. Fig H: Mutational Invariance for Substitutions, 1 bp Insertions, 1 bp Deletions. Each Zimin avoidmer was partitioned into twenty mutually exclusive bins, and for each bin, the invariance potential for each of the mutations was estimated. Fig I: Comparison of STRs in Zimin avoiding regions and genomic regions that do not contain Zimin avoidmers in the human CHM13v2-T2T reference genome. Enrichment represents the STR fold enrichment between the Zimin avoidmer regions and the regions that contain at least one Zimin word. Statistical significance is shown using

Fisher's exact test (p-value<0.001). Fig J: Predicted mutation rates of Zimin avoidmers compared to randomized and GC-Matched controls A. Kernel density estimation of average predicted mutation rate by Roulette across three categories: Avoidmers, Randomized Controls, and GC-Matched Controls. B. Predicted mutation rates across Zimin avoidmers, Randomized Controls, and GC-Matched Controls across three genomic subcompartments, including transcripts, exons, and coding regions. Pairwise comparisons between each control group and their Zimin avoidmers have been conducted using a two-tailed Mann-Whitney test, adjusted for multiple comparisons using Benjamini-Hochberg procedure. Adjusted p-values are displayed as * for p < 0.05, ** for p < 0.01, and *** for p < 0.001. Fig K: Linear regression bubbleplot of CDS to Genome ratio in relationship to Zimin avoidmer density per Mb. Additionally, the bubble size represents the genome-wide GC proportion for each organismal genome.
(DOCX)

**S2 File. Supplementary tables.** Table A: ABA Avoiding Expected Probabilities. Table B: ABACABA Avoiding Expected Probabilities. Table C: Longest Zimin avoidmers for k-mer lengths of 100bps or longer. Table D: Subsets of the most frequent Zimin avoidmers for selected compartments. The two color-highlighted sequences in Hsat1B differ by only 1 bp. Table E: Common Language Effect Size (CLES) of predicted mutation rates between each control group and Zimin avoidmers across three genomic subcompartments. Table F: p-values observed vs. expected probability model organisms, paired sample t-test.
(DOCX)

## Author contributions

**Conceptualization:** Nikol Chantzi, Ilias Georgakopoulos-Soares.

**Data curation:** Nikol Chantzi.

**Formal analysis:** Nikol Chantzi.

**Funding acquisition:** Ilias Georgakopoulos-Soares.

**Investigation:** Nikol Chantzi, Ilias Georgakopoulos-Soares.

**Methodology:** Nikol Chantzi, Ioannis Mouratidis, Ilias Georgakopoulos-Soares.

**Project administration:** Ilias Georgakopoulos-Soares.

**Software:** Nikol Chantzi.

**Supervision:** Ilias Georgakopoulos-Soares.

**Validation:** Nikol Chantzi.

**Visualization:** Nikol Chantzi.

**Writing – original draft:** Nikol Chantzi, Ilias Georgakopoulos-Soares.

**Writing – review & editing:** Nikol Chantzi, Ioannis Mouratidis, Ilias Georgakopoulos-Soares.

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
