## [Decision Letter · Decision Letter 0]

1 Jan 2025

PCOMPBIOL-D-24-01884

Zimin patterns in genomes

PLOS Computational Biology

Dear Dr. Georgakopoulos-Soares,

Thank you for submitting your manuscript to PLOS Computational Biology. After careful consideration, we feel that it has merit but does not fully meet PLOS Computational Biology's publication criteria as it currently stands. Therefore, we invite you to submit a revised version of the manuscript that addresses the points raised during the review process.

Please submit your revised manuscript within 60 days Mar 03 2025 11:59PM. If you will need more time than this to complete your revisions, please reply to this message or contact the journal office at ploscompbiol@plos.org. Please include the following items when submitting your revised manuscript:

We look forward to receiving your revised manuscript.

Kind regards,

Yang Lu, Ph.D.

Academic Editor

PLOS Computational Biology

Jian Ma

Section Editor

PLOS Computational Biology

**Journal Requirements:**

At this stage, the following Authors/Authors require contributions: Nikol Chantzi, Ioannis Mouratidis, and Ilias Georgakopoulos-Soares. Please ensure that the full contributions of each author are acknowledged in the "Add/Edit/Remove Authors" section of our submission form.

5) We notice that your supplementary Figures, and Tables are included in the manuscript file. Please remove them and upload them with the file type 'Supporting Information'. Please ensure that each Supporting Information file has a legend listed in the manuscript after the references list.

6) Please amend your detailed Financial Disclosure statement. This is published with the article. It must therefore be completed in full sentences and contain the exact wording you wish to be published. Please ensure that the funders and grant numbers match between the Financial Disclosure field and the Funding Information tab in your submission form. Note that the funders must be provided in the same order in both places as well.

**Reviewers' comments:**

Reviewer's Responses to Questions

**Comments to the Authors:**

Reviewer #1: In this work, the authors aimed to investigate Zimin words (sequences with identical prefix and suffix) in genomic sequences, specifically exploring "Zimin avoidmers" - sequences that do not contain any Zimin patterns - across multiple organisms. The authors analyzed the Telomere-to-Telomere (T2T) complete human genome as well as other eight model organisms for Zimin words and Zimin avoidmers. In human genome, the authors found that all k-mers above 105 base-pairs contain Zimin words. The Zimin avoidmers have an inhomogeneous distribution in the human genome, and are most enriched in coding sequence (CDS) and human satellite 1 regions. Furthermore, the authors found Zimin avoidmers are associated with loci that have increased sequence diversity in the human genome, but they have a lower insertion and deletion mutation rates. Lastly, the authors identified Zimin avoidmers across model organism genomes and found that Zimin avoidmers are inhomogeneously distributed across model organismal genomes.

The study is well organized and the manuscript is well written. While to study the question of Zimin words in genomes is interesting, I have several concerns and suggestions relating to the biological implications and applications the study.

Major Comments:

1. While the study provides a novel computational approach to analyzing genomic sequences, the broader biological implications are not fully explored. The discussion section hints at potential connections to non-B DNA conformations but it would be better if the authors could expand the discussion on potential applications of Zimin words to synthetic biology and genome studies.

2. The authors note a significant difference between the theoretical maximum Zimin word length and their observed maximum in all the model organisms. While they attribute this to the finite genome size and genome repetitiveness, a more in-depth mathematical and biological explanations of why this discrepancy occurs through mutation process and evolution would strengthen the manuscript.

Minor Comments:

1. Figure 5 is blur and low resolution.

2. Figure 5 legends, there are two B panels.

Reviewer #2: This manuscript investigates the Zimin pattern—where words have identical prefixes and suffixes—in the set of k-mers from the telomere-to-telomere (T2T) human reference genome, as well as reference genomes of eight other model organisms. The authors first explored the pattern of Zimin avoidmers, which are sequences that avoid the Zimin Zn pattern in the T2T genome. They concluded that for k-mer lengths exceeding 104 bp, every k-mer contains a Zimin word. Zimin avoidmers are enriched in coding regions and HSAT1B repetitive regions, and there is a negative correlation between k-mer diversity and the density of the Zimin pattern.

Next, the authors conducted simulations to examine the invariance of Zimin sequences to the introduction of indels and single nucleotide substitutions. They observed a strong enrichment or depletion of Zimin patterns around indels but not substitutions, likely due to nearby short tandem repeats. Finally, they compared Zimin avoidmers across the genomes of model organisms and note that the density of Zimin avoidmers is higher in prokaryotic genomes than in eukaryotic genomes.

Overall, the manuscript is well-written, and the analyses are robust. Below are some minor suggestions and general questions:

1. It would strengthen the manuscript to justify the examination of Zimin patterns in genomes by highlighting the repetitive nature of genomic sequences and presenting a specific hypothesis, rather than stating that “Zimin words have never been examined in bioinformatics.”

2. Figure 3 analysis: Please provide a justification for focusing solely on 8-mers in the analysis presented in Figure 3.

3. In Figure 4C, it would be beneficial to include a comparison of short tandem repeat enrichment in regions that are not Zimin avoidmers.

4. Zimin Words in the Human Genome: What is the pattern of Zimin words (as opposed to Zimin avoidmers) in the human genome?

5. Reverse Complements of Zimin Sequences: Similarly, what is the pattern of the reverse complements of Zimin sequences? Exploring it may further elucidate the characteristics of Zimin patterns but not necessarily to address.

Reviewer #3: The manuscript introduces the concept of Zimin avoidmers and examines their distribution and enrichment across the human reference genome and eight additional model organisms. Key findings include the inhomogeneous distribution of avoidmers, enrichment in coding regions and satellite sequences, and their association with sequence diversity, suggesting potential applications in understanding genomic sequence organization. While the manuscript presents an interesting mathematical concept applied to genomics, several significant concerns prevent me from recommending this submission for acceptance in its current form:

1. While the manuscript introduces an interesting theoretical concept’s applicaiton, it does not sufficiently explore its biological implications or potential applications, leaving the findings somewhat isolated from practical utility for computational biology. Without biological (either from functional or evolutionary) interpretation, this manuscript might be more suitable for journals focusing on computer science or natural language processing.

2. The study assumes that exact matches to Zimin words are meaningful, but this premise lacks biological plausibility. In genomic contexts, spontaneous mutations happen on the genome with certain rate, motifs or secondary structures like hairpin loops are typically error-tolerant and allow for mismatches or variations to some degree. Thus, focusing solely on exact matches to Zimin words is unrealistic and limits the biological relevance of the findings.

3. Statistical rigor is insufficient; there is no statistical significance reported throughout this manuscript. Claims regarding avoidmer distributions, enrichment in genomic compartments, and comparisons across organisms are made without appropriate statistical testing or measures of significance, undermining the reliability of the conclusions.

4. In the section "In the reference human genome every k-mer contains a Zimin word after 104 base-pairs," the statement "with a maximum Zimin avoidmer length of 104 bp emerging on chromosome 17" is inconsistent with Figure 1d, which indicates that chromosome 7 has the maximum avoidmer length.

5. In the section "In the reference human genome every k-mer contains a Zimin word after 104 base-pairs," the authors conclude that "Zimin avoidmers are significantly more GC-rich than the genome average." However, this conclusion should be drawn by comparing the GC content of Zimin avoidmers to that of k-mers containing Zimin words of the same specific length within the genome. Additionally, the authors should provide statistical evidence to support the significance of this difference.

6. In the section "Zimin avoidmers are inhomogeneously distributed across model organismal genomes," the authors claim that Zimin avoidmers are most enriched in genic and CDS regions of the genome. However, this claim is problematic because genic and CDS regions represent only a small fraction of the entire genome. The analysis does not account for the inherent bias introduced by sequence length, as longer sequences are inherently more likely to contain Zimin words due to their size. To make this claim more robust, the authors must normalize for sequence length to accurately assess enrichment and eliminate biases arising from differences in genomic region sizes.

7. In the section “In silico saturation and germline mutagenesis of Zimin avoidmers in the human genome,” the indels are constructed with a specific length of 1 (single-base insertion or deletion), which represents a highly idealized scenario. We suggest extending the analysis to include indels of varying lengths to better reflect real-world genomic mutational processes and evaluate whether the conclusions hold under more biologically realistic conditions.

8. Many of the parameter choices in the study require justification. It is important to assess whether the conclusions remain consistent when varying the bin window length of the genome. Exploring the validity of the findings under different parameter settings is necessary to ensure the robustness and generalizability of the results.

9. The introns, exons, and CDS regions of a gene are not consecutive but rather scattered as separate segments across the genome. It is unclear how Zimin avoidmers are calculated for these fragmented regions. Does the analysis involve concatenating these segments into a single sequence, or are avoidmers identified separately within each segment? This methodological detail is crucial and should be explicitly described to clarify the approach used for Zimin avoidmer identification in these regions.

10. Figures (such as Figure 2) are not aligned between the subplots, and the writing should also be improved.

**Have the authors made all data and (if applicable) computational code underlying the findings in their manuscript fully available?**

Reviewer #1: Yes

Reviewer #2: Yes

Reviewer #3: Yes

PLOS authors have the option to publish the peer review history of their article (what does this mean? ). If published, this will include your full peer review and any attached files.

**Do you want your identity to be public for this peer review?** For information about this choice, including consent withdrawal, please see our Privacy Policy .

Reviewer #1: No

Reviewer #2: No

Reviewer #3: No

**Figure resubmission:**

**Reproducibility:**



---

## [Decision Letter · Decision Letter 1]

19 Sep 2025

PCOMPBIOL-D-24-01884R1

Zimin patterns in genomes

PLOS Computational Biology

Dear Dr. Georgakopoulos-Soares,

Thank you for submitting your manuscript to PLOS Computational Biology. After careful consideration, we feel that it has merit but does not fully meet PLOS Computational Biology's publication criteria as it currently stands. Therefore, we invite you to submit a revised version of the manuscript that addresses the points raised during the review process.

Please submit your revised manuscript within 30 days Nov 19 2025 11:59PM. If you will need more time than this to complete your revisions, please reply to this message or contact the journal office at ploscompbiol@plos.org. Please include the following items when submitting your revised manuscript:

We look forward to receiving your revised manuscript.

Kind regards,

Yang Lu, Ph.D.

Academic Editor

PLOS Computational Biology

Jian Ma

Section Editor

PLOS Computational Biology

**Additional Editor Comments (if provided):**

Reviewer #1:

**Journal Requirements:**

At this stage, the following Authors/Authors require contributions: Nikol Chantzi, Ioannis Mouratidis, and Ilias Georgakopoulos-Soares. Please ensure that the full contributions of each author are acknowledged in the "Add/Edit/Remove Authors" section of our submission form.

2) We have noticed that you have uploaded Supporting Information files, but you have not included a list of legends. Please add a full list of legends for your Supporting Information files after the references list.

3) Please ensure that the funders and grant numbers match between the Financial Disclosure field and the Funding Information tab in your submission form. Note that the funders must be provided in the same order in both places as well.

**Reviewers' comments:**

Reviewer's Responses to Questions

**Comments to the Authors:**

Reviewer #1: Review comments (Remarks to the authors)

Thanks to the authors for their detailed and comprehensive additional analysis to address my comments. Most of my comments have been addressed in this revision. I just have a few more comments based on the authors’ response.

Major:

1.The authors addressed my second comment on mathematical and biological explanations of the Zimin avoidmer lengths. I like the author’s analysis on random permutations of the studied genomes for the mathematical explanations. For biological explanations, in additional to the genome complexity, I’d like to see if the Zimin avoidmer pattern is associated with mutation rates (there are multiple mutation rate models, for example, the authors could find the mutation rates in human genome here: https://www.nature.com/articles/s41588-023-01562-0) or the genome constraint in human genome (for example, the gnocchi score, https://www.nature.com/articles/s41586-023-06045-0). Such analysis would be interesting to broader readers, especially in the human genetics field.

Minor:

1.Figure 5A and 5B captions are mis-matched.

**Have the authors made all data and (if applicable) computational code underlying the findings in their manuscript fully available?**

Reviewer #1: None

PLOS authors have the option to publish the peer review history of their article (what does this mean? ). If published, this will include your full peer review and any attached files.

**Do you want your identity to be public for this peer review?** For information about this choice, including consent withdrawal, please see our Privacy Policy .

Reviewer #1: No

**Figure resubmission:**
---

## [Decision Letter · Decision Letter 2]

9 Jan 2026

Dear Dr. Georgakopoulos-Soares,

We are pleased to inform you that your manuscript 'Zimin patterns in genomes' has been provisionally accepted for publication in PLOS Computational Biology.

Best regards,

Yang Lu, Ph.D.

Academic Editor

PLOS Computational Biology

Jian Ma

Section Editor

PLOS Computational Biology

Reviewer's Responses to Questions

**Comments to the Authors:**

Reviewer #1: Thanks to the authors for addressing my comments on expanding the biological explanations for Zimin avoidermers using mutation rate models. The manuscript is now well written and comprehensive.

**Have the authors made all data and (if applicable) computational code underlying the findings in their manuscript fully available?**

Reviewer #1: Yes

PLOS authors have the option to publish the peer review history of their article (what does this mean? ). If published, this will include your full peer review and any attached files.

**Do you want your identity to be public for this peer review?** For information about this choice, including consent withdrawal, please see our Privacy Policy .

Reviewer #1: No

---

## [Editor Report · Acceptance letter]

PCOMPBIOL-D-24-01884R2

Zimin patterns in genomes

Dear Dr Georgakopoulos-Soares,

I am pleased to inform you that your manuscript has been formally accepted for publication in PLOS Computational Biology. Your manuscript is now with our production department and you will be notified of the publication date in due course.

With kind regards,

Aiswarya Satheesan
